# Stratifying Treatment-Resistant Monosymptomatic Nocturnal Enuresis: Identifying the Subgroup Most Responsive to Biofeedback Therapy

**DOI:** 10.3390/diagnostics15172247

**Published:** 2025-09-05

**Authors:** Emre Kandemir, Ali Sezer, Mehmet Sarikaya

**Affiliations:** 1Department of Urology, Faculty of Medicine, Karamanoglu Mehmetbey University, Karaman 70110, Turkey; emrekandmir@kmu.edu.tr; 2Clinic of Urology and Pediatric Urology, Konya City Hospital, Konya 42100, Turkey; 3Department of Pediatric Surgery, Faculty of Medicine, Selcuk University, Konya 42100, Turkey; drmehmetsarikaya@hotmail.com

**Keywords:** monosymptomatic nocturnal enuresis, biofeedback, bladder capacity, pelvic floor training, treatment resistance

## Abstract

**Background/Objectives**: A subset of children with monosymptomatic nocturnal enuresis (MNE) remains unresponsive to standard treatments such as desmopressin and alarm therapy. This study aimed to identify clinical predictors of response to biofeedback therapy in treatment-resistant MNE and to evaluate the role of bladder capacity as a stratification parameter. **Methods**: In this prospective study, 89 children with treatment-resistant MNE underwent six weekly sessions of biofeedback therapy involving visual pelvic floor feedback. Based on treatment outcomes, patients were classified as complete responders or partial/non-responders. Clinical characteristics including age-adjusted maximal voided volume (MVV), nocturnal polyuria, and wetting frequency were compared. **Results**: Patients with a complete response had significantly lower baseline MVV and age-adjusted MVV (*p* < 0.001). Nocturnal overactivity was more common among responders (60.6% vs. 33.9%, *p* = 0.017), whereas nocturnal polyuria was more frequent in non-responders (*p* = 0.027). Age-adjusted MVV emerged as the only independent predictor of treatment success in multivariate analysis (*p* = 0.045), with ROC analysis confirming its predictive value (AUC = 0.767, 95% CI: 0.667–0.866). **Conclusions**: These findings suggest that reduced bladder capacity and frequent night-time wetting may help identify patients who are more likely to benefit from biofeedback therapy. Bladder capacity assessment may thus serve as a useful tool in tailoring management strategies for refractory MNE.

## 1. Introduction

Enuresis is defined as intermittent and involuntary urinary leakage that occurs during sleep in children over the age of five. Monosymptomatic enuresis (MNE) is characterized by the absence of additional lower urinary tract symptoms (LUTSs), whereas cases presenting with LUTSs are classified as non-monosymptomatic enuresis (NMNE) [1,2,3,4]. MNE affects approximately 5–10% of seven-year-old children and is twice as prevalent in boys. Its etiology is multifactorial, involving genetic, hormonal, psychosocial, developmental, and neurological factors and irregularities in bladder physiology. However, the precise pathophysiology of MNE has not yet been fully elucidated [5,6,7].

One or more potential contributing factors, including nocturnal polyuria, nocturnal bladder overactivity, and heightened sleep arousal, are believed to play a role in the etiology of monosymptomatic nocturnal enuresis (MNE) [8]. The current standard treatment protocol is alarm or desmopressin therapy [9,10]. However, an optimal treatment protocol for patients who do not respond to standard MNE therapy has not yet been established. A comprehensive re-evaluation of potential underlying etiological factors is essential to determine the most effective treatment approach for these patients. Despite the widespread use of desmopressin and alarm therapy, approximately 20–40% of children fail to achieve full dryness, highlighting a clear need for second-line strategies. However, current guidelines lack detailed recommendations for this treatment-resistant group, largely due to insufficient stratification of underlying pathophysiological profiles [10]. Therefore, identification of reliable clinical predictors that may inform treatment selection remains an important unmet need in the management of refractory MNE.

Biofeedback is a therapeutic approach designed to enhance a patient’s awareness of pelvic floor muscle contraction and relaxation through visual, sensory, or tactile stimuli. In recent years, significant advancements have been achieved in treating dysfunctional voiding and urinary incontinence through audiovisual materials, such as computer-assisted animations [11,12,13,14,15,16]. However, biofeedback has not yet been established as an alternative treatment for patients who do not respond to standard therapies. Research suggests that the neural reflex arc responsible for regulating bladder function may be inadequate in children with monosymptomatic nocturnal enuresis (MNE), and biofeedback has the potential to address this deficiency. In this prospective study, we aimed to identify clinical characteristics associated with response to biofeedback therapy in children with treatment-resistant MNE. We hypothesized that reduced bladder capacity and frequent nocturnal wetting would be associated with better response to biofeedback, thus providing a potential framework for individualized therapeutic planning in this challenging patient population.

## 2. Materials and Methods

This prospective study was performed after obtaining approval from the ethics committee. The study was approved by the ethics committee of Karamanoglu Mehmetbey University Faculty of Medicine on 25 May 2021 with the serial number 03/2021-01. Written informed consent was obtained from the legal descendants of all participants. The study’s steps were planned and performed according to the Helsinki Declaration of the World Medical Association. This prospective study included patients who were admitted due to MNE between June 2021 and July 2024, were between the ages of 6–14, had enuresis three or more times a week for six consecutive months, and did not respond to regular desmopressin and alarm treatment for three months. A priori power analysis was not performed; however, the sample size was deemed sufficient based on the available patient population and similar prior studies. Patients with LUTSs, such as daytime incontinence, dysuria, holding maneuvers, and urgency, were excluded from the study. Patients who had respiratory tract obstruction, learning difficulties, perceptual, auditory, or visual problems, a history of pelvic surgery, and who had not fully cooperated with behavioral therapy before were excluded. A detailed history and physical examination were performed on all patients. Anal sphincter tone, bulbocavernous reflex, and perineal sensitivity were evaluated in all patients. Urinalysis, ultrasonography (USG), uroflowmetry (UFM) with electromyography (EMG), a 2-day daytime bladder diary, and at least one week of enuresis charts were noted. Standard urotherapy (behavioral modifications) was offered to all patients as an initial step. Voiding posture and toilet training were given. Bladder irritants or liquids with diuretic effects (containing caffeine, etc.) were removed from the diet. Fluid restriction was advised for two hours before sleep.

Patients’ age, gender, family history (mother, father, both parents), body mass index, frequency of weekly incontinence (1–2, 3–5, 6–7 night incontinence per week), presence of nocturnal polyuria (diaper weighing at night + first urine in the morning ≥ 130% expected bladder capacity) and nocturnal overactivity (≥1-time enuresis per night) were recorded.

Biofeedback therapy was administered using the MMS 500 system (MMS International, Enschede, The Netherlands), incorporating uroflowmetry and surface electromyography (UFM-EMG). Two perineal surface electrodes were placed at 3 and 9 o’clock positions, and a reference electrode was attached to the left knee. Visual feedback was provided through a computer-assisted animation (Gymna Mno game), in which children were instructed to contract and relax pelvic floor muscles to control the movement of animated birds.

Each session lasted 30 min and was conducted once per week for six consecutive weeks. During the sessions, patients performed 10–15 cycles of pelvic floor muscle contraction and relaxation under supervision. Between sessions, patients were instructed to perform home exercises consisting of three sets of 10 contractions per day, with each contraction held for five seconds, followed by five seconds of rest. The same trained pediatric nurse conducted all sessions to ensure consistency

Bladder capacity and maximal voided volume (MVV) were recorded before and after treatment. Age-adjusted MVV was calculated by dividing MVV by expected bladder capacity ((age + 1)*30) [17]. Clinical response was evaluated due to the ICCS criteria. A decrease of less than 50% in the number of wet days was interpreted as nonresponse, 50–99% as a partial response, and 100% as a complete response.

After six biofeedback therapy sessions, patients were divided into two groups: complete response and nonresponse or partial response. Two groups were compared in terms of age, gender, family history, body mass index, weekly leakage frequency, presence of nocturnal polyuria, presence of nocturnal overactivity, change in USG and UFM values, voided volumes, post-void residual (PVR) values, and Qmax values (pre-treatment and post-treatment).

Statistical analyses were performed using SPSS version 22.0. The distribution of continuous variables was assessed using the Shapiro–Wilk test. Between-group comparisons were made using the independent samples *t*-test or the Mann–Whitney U test, depending on data normality. Categorical variables were analyzed using the Chi-square test or Fisher’s exact test, as appropriate. Pre- and post-treatment changes in continuous variables (e.g., Qmax, bladder capacity, PVR) were evaluated using the Wilcoxon signed-rank test. A multivariate logistic regression analysis was performed to identify independent predictors of treatment response. A *p*-value of <0.05 was considered statistically significant. To assess the discriminative ability of age-adjusted bladder capacity in predicting treatment response, receiver operating characteristic (ROC) curve analysis was performed. The area under the curve (AUC), sensitivity, specificity, and Youden’s index were calculated. The optimal cutoff value was determined based on the maximum Youden’s index.

## 3. Results

A total of 112 patients with refractory MNE were evaluated for the study. Eleven patients who did not meet the study criteria were excluded, and eight had lost follow-up. Four patients declined participation. Eighty-nine patients were included in the study. The flow diagram is shown in Figure 1.

The mean age of 89 treatment-resistant patients was 8.5 ± 2 years. 58 (65.2%) patients were male, and 31(34.8%) were female. Positive family history was present in 76.4% of the patients. In 52 (58.4%) patients, only 1 parent had a history of enuresis, and in 16 (18%) patients, both parents had a history of enuresis. There was no statistically significant difference between the groups regarding family history (*p* = 0.867). The average body mass index was 19.9 ± 4.9 kg/m^2^; no significant difference was detected between the groups (*p* = 0.438).

The number of wet nights per month was 18.3 ± 6.1 before treatment, and it decreased to 6.2 ± 5.6 after treatment (group 1: 0.9 ± 0.6, group 2: 9.4 ± 4.8, *p* = <0.001). The maximal voided volume (MVV) before biofeedback treatment was 194 ± 47.1 mL in group 1 and 237.2 ± 62.9 mL in group 2, and the difference was statistically significant (*p* = 0.001). Although MVV increased in both groups after biofeedback treatment, the difference was insignificant (*p* = 0.465). Age-adjusted MVV was significantly lower in group 1 before treatment (68.8% ± 10.1 and 83.1% ± 17.1, respectively, *p* < 0.001). In terms of nocturnal polyuria, it was statistically significantly higher in group 2 (*p* = 0.027). Nocturnal overactivity (enuresis > 1 per night) was 60.6% in group 1 and 33.9% in group 2, and the difference was statistically significant (*p* = 0.017). Significant differences between groups are shown in Figure 2. In the multivariate analysis, only age-adjusted MVV was statistically significant (*p* = 0.045).

The patients’ pre- and post-treatment data are given in Table 1. Multivariate logistic regression analysis was performed to determine independent predictors of complete response to biofeedback therapy. Nocturnal polyuria was found to be significantly associated with a lower likelihood of response (OR: 0.34; 95% CI: 0.12–0.94; *p* = 0.039). Age-adjusted maximal voided volume (%) was also identified as a significant predictor (OR: 0.97; 95% CI: 0.94–0.99; *p* = 0.017). Wet nights per week (OR: 0.92; 95% CI: 0.77–1.09; *p* = 0.33) and age (OR: 0.89; 95% CI: 0.70–1.14; *p* = 0.37) were not statistically significant (Table 2). ROC curve analysis demonstrated that age-adjusted bladder capacity had acceptable discriminative power for predicting treatment response, with an area under the curve (AUC) of 0.767 (95% CI: 0.667–0.866, *p* < 0.001). At the optimal cutoff value (68.1%), sensitivity was 54.5% and specificity was 75.0%. (Figure 3).

## 4. Discussion

Modern treatment methods for enuresis can be traced back to the 19th century. In the earliest recorded instances of the disease, chloral hydrate, a sedative and hypnotic drug, was utilized as a treatment for enuresis in 1871 [18]. In the latter half of the 20th century, alarm therapy, imipramine, and desmopressin were introduced as established treatment options for enuresis, in that order [19,20,21,22,23,24,25,26]. There has been no significant change in the standard treatment of MNE in the last 40 years. The optimal approach for treatment-resistant cases remains unclear, mainly due to the incomplete understanding of the disease’s etiology and pathophysiology. Currently, desmopressin and alarm therapy remain the standard treatments for MNE. However, alternative therapeutic options are still required for patients who do not respond to these standard interventions. While daytime LUTSs are not typically observed in MNE patients, nocturnal polyuria and bladder overactivity may be present in treatment-resistant individuals. Stratifying patients based on underlying pathophysiological features may facilitate more targeted and cost-effective management approaches. To our knowledge, this is the first study to evaluate the predictive role of age-adjusted bladder capacity and nocturnal polyuria in determining response to biofeedback therapy in a strictly defined cohort of treatment-resistant monosymptomatic enuresis patients.

Biofeedback treatment is a neuromodulatory intervention designed to improve lower urinary tract function by enhancing awareness and voluntary control of pelvic floor muscles. It has shown efficacy in managing pediatric conditions such as overactive bladder, high post-void residual (PVR), and dysfunctional voiding syndromes, including spinning top urethra [14]. This technique regulates neuromuscular coordination by modulating both sphincteric hypoexcitability and detrusor hyperexcitability [27]. Despite its established role in other pediatric urological conditions, the utility of biofeedback in the treatment of monosymptomatic nocturnal enuresis (MNE) remains unclear. Although desmopressin and alarm therapy show similar efficacy rates (60–70%) in standard cases, a subset of patients remains refractory [28]. In our cohort, biofeedback demonstrated clinical benefit in a proportion of these resistant patients, prompting the need to identify which specific clinical features predict a favorable response.

In our study, patients who responded to biofeedback therapy exhibited significantly lower baseline maximal voided volume (MVV) and age-adjusted MVV compared to non-responders. This finding aligns with previous research indicating that diminished functional bladder capacity is commonly observed in children with refractory enuresis who fail to respond to desmopressin, alarm, or anticholinergic therapies [29]. Importantly, both MVV and age-adjusted MVV improved following biofeedback intervention in our series. This suggests that biofeedback may aid in modulating bladder capacity by enhancing neuromuscular control and awareness of pelvic floor function. Given its use of real-time visual feedback to modulate skeletal muscle activity, biofeedback may also exert central effects by promoting brain–bladder coordination, a mechanism that warrants further investigation.

The predictive value of age-adjusted bladder capacity was further supported by receiver operating characteristic (ROC) analysis, which demonstrated an area under the curve (AUC) of 0.767. This finding indicates an acceptable level of discriminative power for identifying patients who are more likely to respond to biofeedback therapy. At the optimal cutoff value (68.1%), the sensitivity and specificity were 54.5% and 75.0%, respectively. These results reinforce the clinical utility of bladder capacity measurements not only as a descriptive parameter but also as a predictive tool in treatment planning. Incorporating such objective metrics into routine assessment may help guide individualized therapy decisions and improve outcomes in treatment-resistant MNE.

Although decreased functional bladder capacity (FBC) is more commonly associated with non-monosymptomatic enuresis (NMNE), emerging evidence suggests that it is not uncommon among patients with monosymptomatic nocturnal enuresis (MNE) as well. This has challenged the earlier assumption that bladder capacity remains normal in MNE. Liu et al. reported that 33.9% of children with MNE demonstrated reduced FBC despite the absence of daytime symptoms [30], while Baek et al. found a similar prevalence of 27.8% in newly diagnosed cases [31]. These findings underscore the importance of evaluating bladder capacity—commonly estimated via maximal voided volume (MVV)—as a potential indicator of disease severity and treatment responsiveness. Our study reinforces this perspective by demonstrating a clear association between reduced MVV and positive response to biofeedback therapy.

Our findings indicate that biofeedback therapy was significantly less effective in patients with nocturnal polyuria, as the response rate in this subgroup was notably lower compared to those without nocturnal polyuria. This observation aligns with existing literature suggesting that nocturnal polyuria is often linked to insufficient nocturnal vasopressin secretion, a mechanism not addressed by pelvic floor-based interventions such as biofeedback [29,32]. Since the therapeutic effect of biofeedback appears to rely on improving bladder capacity and neuromuscular control, it may have limited utility in cases where excessive nocturnal urine production is the primary pathology. Given these findings, clinicians may consider optimizing fluid intake strategies or adjusting desmopressin dosage in patients with prominent NP prior to initiating biofeedback therapy.

Our study is among the first to investigate the predictive value of wetting frequency per night on the effectiveness of biofeedback therapy in MNE. Interestingly, patients who experienced multiple wetting episodes per night were more likely to respond to treatment, with a response rate of 60.6% in this subgroup compared to 33.9% among those with single wetting episodes (*p* = 0.017). This finding may reflect a distinct pathophysiological profile characterized by nocturnal bladder overactivity, which could be more amenable to neuromodulatory interventions such as biofeedback.

In multivariate logistic regression analysis, nocturnal polyuria and age-adjusted maximal voided volume (MVV) emerged as independent predictors of treatment response to biofeedback therapy. This supports the hypothesis that bladder capacity and nocturnal urine production represent distinct pathophysiological dimensions influencing treatment outcomes. The lack of significance for age and wetting frequency further suggests that urodynamic and functional bladder parameters may be more relevant than demographic or symptomatic indicators. Specifically, the presence of nocturnal polyuria was associated with a significantly lower likelihood of complete response, supporting previous observations that excessive nocturnal urine production represents a pathophysiological mechanism not targeted by pelvic floor-based interventions. Furthermore, lower age-adjusted MVV values independently predicted better treatment outcomes, suggesting that patients with reduced functional bladder capacity may benefit more from bladder control training. These findings underscore the importance of patient stratification in guiding treatment decisions and highlight the potential utility of bladder capacity assessments and nocturnal urine profiling in optimizing the selection of candidates for biofeedback therapy. Our findings support a shift toward precision medicine in the management of treatment-resistant MNE, whereby objective functional markers such as age-adjusted bladder capacity may help identify children most likely to benefit from biofeedback therapy. Rather than applying a uniform approach to all refractory cases, a stratified model incorporating bladder capacity metrics, nocturnal polyuria assessment, and wetting frequency per night could guide clinical decision-making. Such an approach not only enhances therapeutic efficiency but may also reduce unnecessary treatment burdens for non-responders. The integration of simple, non-invasive tools like bladder diaries and adjusted MVV calculations into routine evaluation could serve as cost-effective biomarkers in tailoring neuromodulatory interventions. Further prospective studies with larger cohorts are warranted to validate these stratification criteria and to explore whether combined strategies (e.g., biofeedback plus desmopressin in selected cases) can enhance response rates. These findings may aid clinicians in selecting candidates for biofeedback therapy by focusing on patients with reduced bladder capacity and without significant nocturnal polyuria, thereby improving the efficiency of individualized treatment planning.

While patients with MNE typically lack daytime lower urinary tract symptoms, nocturnal bladder overactivity may underlie persistent wetting episodes in a subset of cases. In such situations, anticholinergic agents are commonly employed; however, their efficacy and tolerability in pediatric populations remain variable. Our findings suggest that biofeedback therapy may serve as a non-pharmacologic alternative in this subgroup, potentially enhancing central neuromodulation and brain–bladder coordination. Given its established role in managing daytime overactive bladder, biofeedback may exert similar regulatory effects during sleep. For patients unresponsive to first-line therapies and exhibiting signs of nocturnal bladder overactivity, biofeedback represents a promising adjunct or alternative treatment strategy. A key strength of this study lies in its attempt to stratify treatment-resistant MNE patients using functional bladder parameters and nocturnal voiding characteristics. By identifying age-adjusted bladder capacity and nocturnal polyuria as independent predictors of biofeedback response, our findings contribute to the development of a more individualized and physiologically informed treatment approach. To our knowledge, this is among the first studies to provide a targeted stratification framework for biofeedback therapy in this specific patient population.

While some previous studies have reported no significant benefit of biofeedback in MNE management, a key limitation in many of these trials is the lack of patient stratification. For instance, a study involving 63 children incorporated biofeedback into standard treatment protocols but analyzed the cohort as a single, homogeneous group [33]. In contrast, our study revealed that biofeedback was effective in 37.1% of treatment-resistant patients, and subgroup analysis uncovered distinct clinical patterns associated with treatment responsiveness. These findings highlight the potential limitations of generalized treatment approaches and support the use of individualized strategies—particularly targeting patients with signs of nocturnal bladder overactivity—for optimizing therapeutic outcomes. Although our study demonstrated a significant short-term response to biofeedback therapy in a subset of patients with treatment-resistant MNE, the long-term sustainability of this therapeutic benefit remains unknown. Future studies incorporating extended follow-up periods are necessary to determine whether the observed improvements persist over time or diminish after the cessation of therapy. Assessing relapse rates and long-term adherence may provide further insight into the durability and practical utility of biofeedback interventions in clinical practice.

This study has several limitations that should be considered when interpreting the findings. First, the definition of complete response was based on parental reports rather than objective bladder diaries or urodynamic measurements, which may introduce reporting bias. Second, the relatively short follow-up period limits the ability to assess long-term treatment durability. Third, although we performed multivariate logistic regression to adjust for potential confounders, unmeasured factors such as sleep quality, psychological comorbidities, and adherence to biofeedback protocols may have influenced treatment outcomes. Additionally, the sample size, particularly within subgroups such as those with nocturnal polyuria, may have limited the statistical power to detect subtle associations.

## 5. Conclusions

MNE remains a common pediatric condition, with a substantial proportion of patients exhibiting resistance to standard therapies such as desmopressin and alarm treatment. Our findings suggest that stratifying patients based on functional bladder capacity and wetting frequency may enhance treatment personalization and improve clinical outcomes. Biofeedback therapy, as a non-invasive and child-friendly intervention, appears to be particularly effective in patients with reduced MVV, lower age-adjusted bladder capacity, and frequent nocturnal wetting. These results highlight the potential value of incorporating bladder capacity metrics into routine assessment for treatment-resistant MNE. Further prospective, controlled studies are needed to validate these findings and define the optimal role of biofeedback within individualized management strategies.

## Figures and Tables

**Figure 1 diagnostics-15-02247-f001:**
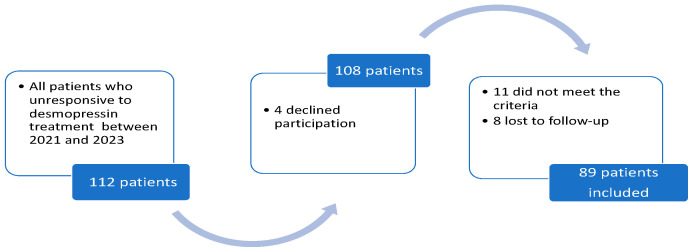
Flowchart of patient inclusion and treatment response classification.

**Figure 2 diagnostics-15-02247-f002:**
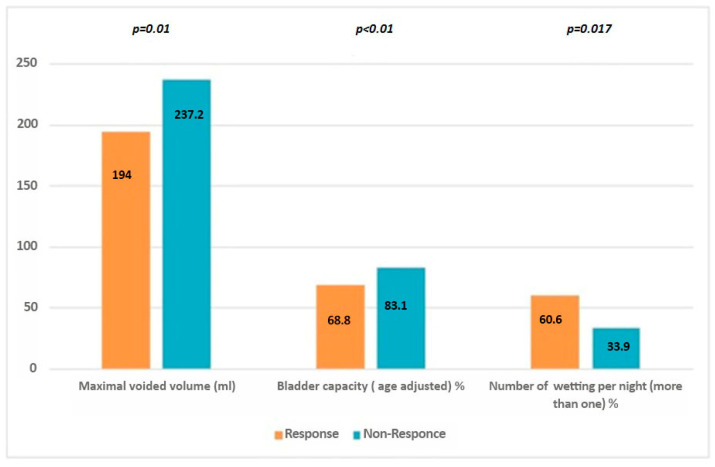
Distribution of age-adjusted maximal voided volume in responders and non-responders.

**Figure 3 diagnostics-15-02247-f003:**
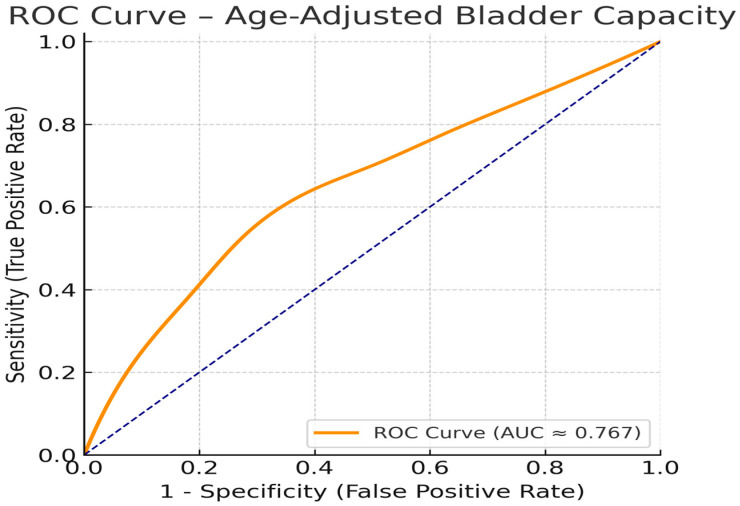
ROC curve assessing the diagnostic value of age-adjusted bladder capacity in predicting complete response to biofeedback therapy. The analysis yielded an AUC of 0.767 (95% CI: 0.667–0.866), with an optimal cutoff of 68.1% corresponding to a sensitivity of 54.5% and specificity of 75.0%.

**Table 1 diagnostics-15-02247-t001:** Evaluation of treatment results in terms of demographic and clinical parameters.

	Overall	Response (Group 1)	Nonresponse (Group 2)	*p*	Multivariant
**Patient**, *n* (%)	89	33 (37.1%)	56 (62.9%)		
**Gender**, *n* (%)				0.646	
Male Female	58 65.2%) 31 (34.8%)	23 10	35 21		
**Family history**, *n* (%)				0.867	
None One parent Both parents	21 (23.6%) 52 (58.4%) 16 (18%)	8 20 5	13 32 11		
**Age** (year) *****	8.5 ± 2	8.4 ± 2.1	8.6 ± 2.0	0.736	
**Body mass index** (kg/m^2^) *****	19.9 ± 4.9	20.5 ± 5.1	19.7 ± 4.8	0.438	
**Maximal voided volume** *****					
Before biofeedback therapy After biofeedback therapy	221.2 ± 61 246.4 ± 60.6	194 ± 47.1 240.2 ± 56.6	237.2 ± 62.9 250 ± 63.1	<0.001 0.465	
**Age-adjusted maximal voided volume** %					
Before biofeedback therapy After biofeedback therapy	77.8 ± 16.4 87.1 ± 16.8	68.8 ± 10.1 85.9 ± 15.9	83.1 ± 17.1 87.8 ± 17.5	<0.001 0.618	0.045 **^β^**
**Number of wet nights (per month) ***					
Before biofeedback therapy After biofeedback therapy	18.3 ± 6.1 6.2 ± 5.6	17.4 ± 6.1 0.9 ± 0.6	18.8 ± 6.0 9.4 ± 4.8	0.344 <0.001	
**Nocturnal poliuria**, *n* (%)				0.027	
Yes No	13 (14.6%) 76 (85.4%)	1 32	12 44		
**Number of wettings per night**, *n* (%)				0.017	
One More than one	50 (56.2) 39 (43.8%)	13 20	37 19		

* mean ± standard deviation. **^β^**: OR:0.83%95 GA 0.66–0.86.

**Table 2 diagnostics-15-02247-t002:** Multivariate logistic regression analysis for predictors of complete response to biofeedback therapy.

	Odds Ratio (OR)	95% CI	*p*-Value
**Nocturnal polyuria (yes vs. no)**	0.34	0.12–0.94	0.039
**Age-adjusted MVV (%)**	0.97	0.94–0.99	0.017
**Wet nights/week**	0.92	0.77–1.09	0.33
**Age (years)**	0.89	0.70–1.14	0.37

## Data Availability

The data presented in this study are available on request from the corresponding author due to legal issues.

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
