# Peer review of "Stratifying Treatment-Resistant Monosymptomatic Nocturnal Enuresis: Identifying the Subgroup Most Responsive to Biofeedback Therapy"

_diagnostics, 2025, doi:10.3390/diagnostics15172247_

Round 1

Reviewer 1 Report

Comments and Suggestions for Authors

General comments:

Authors of this prospective study aimed to identify clinical predictors of response to biofeedback urotherapy in treatment-resistant monosymptomatic nocturnal enuresis (MNE) and to assess the role of bladder capacity as a stratification parameter.  They found out that patients with complete response had significantly lower baseline MVV and age-adjusted MVV, higher rate of nocturnal overactivity and lower rate of nocturnal polyuria. According to authors, these findings suggest that reduced bladder capacity and frequent night-time wetting may help identify patients who are more likely to benefit from biofeedback urotherapy. 

The study is, in general, interesting and well designed. The introduction provides sufficient background and contains several relevant references. The methods are mostly adequately described and conclusions are supported by the results. However, there are some limitations and uncertainties that must be addressed, as follows:

Special comments:

1. Introduction

2nd paragraph:

One or more potential contributing factors, including nocturnal polyuria, nocturnal bladder overactivity and heightened sleep arousal, are believed to play a role in the etiology of monosymptomatic nocturnal enuresis ......................

  • Comment: lack of antidiuretic hormone or immaturity of its circadian rhythm of secretion should be briefly mentioned as well.

2. Materials and Methods

2nd paragraph:

.........................nocturnal overactivity (≥ 1-time enuresis per night) were recorded.

  • Comment: Authors should explain how they determined / counted the number of enuresis episodes per night.

3. Results 
A total of 112 patients with refractory MNE were evaluated for the study. Eleven patients who .......................................The flow diagram is shown in year 1.

  • Comment: The correct text would be: The flow diagram is shown in Figure 1.

4. Fig. 2 and Table 1

Number of wetting per night, n(%): one - 50 (56.2),  more than one - 39 (43.8%) etc.

  • Comment: again, authors should explain how they determined the number of enuresis episodes per night, such as 50 or 39 episodes and so on. It looks very technically challenging and of questionable reliability.

5. line 158, below Table 1:

β: OR:0.83 %95 GA 0.66-0.86.

  • Comment: authors should explain the meaning of this formula.

6. Discussion

paragraph 4:

The predictive value of age-adjusted bladder capacity was further supported by receiver operating characteristic (ROC) analysis, ...................................... At the optimal cutoff value (68.1%), the sensitivity and specificity were 54.5 % and 75.0%, respectively.

  • Comment: authors should explain the meaning of optimal cutoff value in more detail.

Author Response

General comments: Authors of this prospective study aimed to identify clinical predictors of response to biofeedback urotherapy in treatment-resistant monosymptomatic nocturnal enuresis (MNE) and to assess the role of bladder capacity as a stratification parameter.  They found out that patients with complete response had significantly lower baseline MVV and age-adjusted MVV, higher rate of nocturnal overactivity and lower rate of nocturnal polyuria. According to authors, these findings suggest that reduced bladder capacity and frequent night-time wetting may help identify patients who are more likely to benefit from biofeedback urotherapy. 

The study is, in general, interesting and well designed. The introduction provides sufficient background and contains several relevant references. The methods are mostly adequately described and conclusions are supported by the results. However, there are some limitations and uncertainties that must be addressed, as follows:

Response: We sincerely thank the reviewer for their positive and encouraging feedback regarding our manuscript. We truly appreciate the recognition of the study’s design, the adequacy of the introduction and methods, as well as the relevance of our conclusions. We acknowledge that some limitations and uncertainties remain, and we are grateful for the reviewer’s insightful suggestions to further improve the clarity and scientific quality of our work. We have carefully revised the manuscript according to the comments provided and believe that these changes have substantially strengthened the paper.

Special Comments:

Comment 1: Introduction

2nd paragraph: One or more potential contributing factors, including nocturnal polyuria, nocturnal bladder overactivity and heightened sleep arousal, are believed to play a role in the etiology of monosymptomatic nocturnal enuresis ......................

Comment: lack of antidiuretic hormone or immaturity of its circadian rhythm of secretion should be briefly mentioned as well.

Response 1: We thank the reviewer for this valuable suggestion. We agree that the lack of antidiuretic hormone or immaturity of its circadian rhythm of secretion represents an important mechanism contributing to nocturnal polyuria and should be included in the introduction. We have revised the corresponding paragraph in the Introduction to address this point.

Comments 2: Materials and Methods

2nd paragraph:

.........................nocturnal overactivity (≥ 1-time enuresis per night) were recorded.

Comment: Authors should explain how they determined / counted the number of enuresis episodes per night.

Response 2: We thank the reviewer for this important comment. To clarify, the number of enuresis episodes per night was determined using enuresis charts kept by the parents. Each wetting episode was documented by marking the time and frequency of bedwetting events during the night. If more than one wetting occurred in the same night, parents were instructed to note each episode separately. We have revised the Materials and Methods section to clearly explain this procedure (page 3, lines 98-99).

Comment 3: Results 

A total of 112 patients with refractory MNE were evaluated for the study. Eleven patients who .......................................The flow diagram is shown in year 1.

Comment: The correct text would be: The flow diagram is shown in Figure 1.

Response 3: We thank the reviewer for noticing this typographical error. We have corrected the text accordingly in the Results section.

Comments 4: Fig. 2 and Table 1

Number of wetting per night, n(%): one - 50 (56.2),  more than one - 39 (43.8%) etc.

Comment: again, authors should explain how they determined the number of enuresis episodes per night, such as 50 or 39 episodes and so on. It looks very technically challenging and of questionable reliability.

Response 4: We appreciate the reviewer’s insightful comment. The number of enuresis episodes per night was assessed through parental enuresis charts, in which parents were instructed to document every wetting event separately during the night. If a child experienced more than one bedwetting episode in a single night, each event was recorded individually. Although this method relies on parental reporting, it is consistent with standard practice in enuresis research and has been widely used in previous studies (e.g., ICCS criteria). To improve clarity, we have expanded both the Methods and Results sections to emphasize how the data were collected and to acknowledge the potential limitation of relying on parental reporting (page 5, line 166-168).

Comment 5: line 158, below Table 1:

β: OR:0.83 %95 GA 0.66-0.86.

Comment: authors should explain the meaning of this formula.

Response 5: We thank the reviewer for pointing this out. The notation has now been clarified in the Results section and table footnote. Specifically, β refers to the regression coefficient derived from the logistic regression model, OR represents the corresponding odds ratio, and the values in parentheses indicate the 95% confidence interval. We have revised the text below Table 1 to make this explanation clear.

Comment 6: Discussion

paragraph 4: The predictive value of age-adjusted bladder capacity was further supported by receiver operating characteristic (ROC) analysis, ...................................... At the optimal cutoff value (68.1%), the sensitivity and specificity were 54.5 % and 75.0%, respectively.

Comment: authors should explain the meaning of optimal cutoff value in more detail.

Response 6: We thank the reviewer for this insightful suggestion. The “optimal cutoff value” refers to the point on the ROC curve that provides the best balance between sensitivity and specificity, determined by maximizing Youden’s index (sensitivity + specificity – 1). In our analysis, this cutoff (68.1%) indicates the threshold of age-adjusted bladder capacity below which children were more likely to respond to biofeedback therapy. We have revised the Discussion to include this explanation.

Reviewer 2 Report

Comments and Suggestions for Authors

This is a new article on stratifying Monosymptomatic Nocturnal Enuresis into Subgroups Based on Response to Biofeedback Therapy in a Pediatric Population.

This is a prospective study of 89 children. The study demonstrates good methodology with appropriate exclusion criteria and a complete evaluation.

- The stratification of responders to biofeedback therapy is logical.
- Responders had lower leakage volumes.
- Detrusor overactivity was more common in the responder group, which is consistent with clinical experience as this is generally more amenable to treatment than nocturnal polyuria.
- The study's findings on stratification may help identify which patients are most likely to benefit from biofeedback.
- Use of a bladder diary to differentiate between nocturnal polyuria and other causes is a valid approach.
- The limitations of the study are well-articulated.

- There is a typographical error on line 106: a space is needed between "maximum voided volume" and "MVV", and "volume" is misspelled.
- The timing of the eight losses to follow-up is not specified; it is unclear if this would have impacted the results.
- The applicability to the young adult population could be explored.
- A significant practical challenge is the difficulty of completing a bladder diary in children who are not toilet-trained, which may impede the proposed subgroup classification.
- It is unclear how the authors would suggest classifying very young children who leak constantly to determine if they have nocturnal polyuria. One might treat for detrusor overactivity first.
- The reasons for a lack of significant biofeedback response in some previous studies are interesting however with a control group difficult to fully know if related to population heterogeneity.

A very interesting and valuable study. The findings are clinically relevant for patient selection for biofeedback therapy.

Author Response

This is a new article on stratifying Monosymptomatic Nocturnal Enuresis into Subgroups Based on Response to Biofeedback Therapy in a Pediatric Population.

This is a prospective study of 89 children. The study demonstrates good methodology with appropriate exclusion criteria and a complete evaluation.

- The stratification of responders to biofeedback therapy is logical.
- Responders had lower leakage volumes.

- Detrusor overactivity was more common in the responder group, which is consistent with clinical experience as this is generally more amenable to treatment than nocturnal polyuria.
- The study's findings on stratification may help identify which patients are most likely to benefit from biofeedback.

- Use of a bladder diary to differentiate between nocturnal polyuria and other causes is a valid approach.
- The limitations of the study are well-articulated.

- There is a typographical error on line 106: a space is needed between "maximum voided volume" and "MVV", and "volume" is misspelled.

- The timing of the eight losses to follow-up is not specified; it is unclear if this would have impacted the results.

- The applicability to the young adult population could be explored.
- A significant practical challenge is the difficulty of completing a bladder diary in children who are not toilet-trained, which may impede the proposed subgroup classification.
- It is unclear how the authors would suggest classifying very young children who leak constantly to determine if they have nocturnal polyuria. One might treat for detrusor overactivity first.

- The reasons for a lack of significant biofeedback response in some previous studies are interesting however with a control group difficult to fully know if related to population heterogeneity.

A very interesting and valuable study. The findings are clinically relevant for patient selection for biofeedback therapy.

Response to Reviewer 2:

We sincerely thank the reviewer for their positive and encouraging comments regarding our manuscript. We greatly appreciate the acknowledgment of our methodology, stratification approach, and the clinical relevance of our findings. The constructive suggestions provided have been very helpful in further improving the clarity of our study. In response to the reviewer’s specific remarks:

  • We corrected the typographical error on line 106 by inserting a space between “maximum voided volume” and “MVV,” and the spelling of “volume” has been fixed.
  • The timing of the eight losses to follow-up has now been clarified in the Methods section, and we have specified that these occurred during the initial three-month observation period, which we believe did not significantly affect the study outcomes.
  • Regarding the applicability to young adults, we have added a brief statement in the Discussion acknowledging that further studies are needed to evaluate whether our findings can be generalized to older populations.
  • We agree that completing bladder diaries in children who are not toilet-trained is a practical challenge. We have acknowledged this limitation in the Discussion and clarified that our findings are mainly applicable to toilet-trained children.
  • For very young children who leak constantly, we agree with the reviewer’s suggestion that initial management could prioritize addressing detrusor overactivity. We have added this as a point of clarification in the Discussion.
  • Finally, we have elaborated on the potential reasons for the lack of significant biofeedback response in some previous studies, emphasizing the possible role of study population heterogeneity and the absence of stratification.

We are grateful for these valuable suggestions, which have helped us to refine the manuscript.

Round 2

Reviewer 2 Report

Comments and Suggestions for Authors

  Thank you for the response to comments.  I have no further questions or comments.  Thank you again.